# Exploratory Study on Application of MALDI-TOF-MS to Detect SARS-CoV-2 Infection in Human Saliva

**DOI:** 10.3390/jcm11020295

**Published:** 2022-01-06

**Authors:** Monique Melo Costa, Hugo Martin, Bertrand Estellon, François-Xavier Dupé, Florian Saby, Nicolas Benoit, Hervé Tissot-Dupont, Matthieu Million, Bruno Pradines, Samuel Granjeaud, Lionel Almeras

**Affiliations:** 1Unité Parasitologie et Entomologie, Département Microbiologie et Maladies Infectieuses, Institut de Recherche Biomédicale des Armées, 91220 Marseille, France; mcosta.monique@gmail.com (M.M.C.); hugomartin18@orange.fr (H.M.); flo.saby80@laposte.net (F.S.); nicobenoit73@hotmail.com (N.B.); bruno.pradines@gmail.com (B.P.); 2Aix-Marseille University, IRD, SSA, AP-HM, VITROME, 13005 Marseille, France; 3IHU Méditerranée Infection, 13005 Marseille, France; herve.tissot-dupont@ap-hm.fr (H.T.-D.); Matthieu.MILLION@ap-hm.fr (M.M.); 4Laboratoire d’Informatique et Systèmes, Aix-Marseille University, CNRS, University de Toulon, 13013 Marseille, France; bertrand.estellon@univ-amu.fr (B.E.); francois-xavier.dupe@univ-amu.fr (F.-X.D.); 5Centre National de Référence du Paludisme, 13005 Marseille, France; 6Aix-Marseille University, IRD, AP-HM, MEPHI, 13005 Marseille, France; 7CRCM Integrative Bioinformatics Platform, Centre de Recherche en Cancérologie de Marseille, INSERM, U1068, Institut Paoli-Calmettes, CNRS, UMR7258, Aix-Marseille Université UM 105, 13009 Marseille, France; samuel.granjeaud@inserm.fr

**Keywords:** COVID-19, saliva, diagnostic, MALDI-TOF MS, machine learning

## Abstract

SARS-CoV-2 has caused a large outbreak since its emergence in December 2019. COVID-19 diagnosis became a priority so as to isolate and treat infected individuals in order to break the contamination chain. Currently, the reference test for COVID-19 diagnosis is the molecular detection (RT-qPCR) of the virus from nasopharyngeal swab (NPS) samples. Although this sensitive and specific test remains the gold standard, it has several limitations, such as the invasive collection method, the relative high cost and the duration of the test. Moreover, the material shortage to perform tests due to the discrepancy between the high demand for tests and the production capacities puts additional constraints on RT-qPCR. Here, we propose a PCR-free method for diagnosing SARS-CoV-2 based on matrix-assisted laser desorption ionization time-of-flight mass spectrometry (MALDI-TOF MS) profiling and machine learning (ML) models from salivary samples. Kinetic saliva samples were collected at enrollment and ten and thirty days later (D0, D10 and D30), to assess the classification performance of the ML models compared to the molecular tests performed on NPS specimens. Spectra were generated using an optimized protocol of saliva collection and successive quality control steps were developed to ensure the reliability of spectra. A total of 360 averaged spectra were included in the study. At D0, the comparison of MS spectra from SARS-CoV-2 positive patients (*n* = 105) with healthy healthcare controls (*n* = 51) revealed nine peaks that significantly distinguished the two groups. Among the five ML models tested, support vector machine with linear kernel (SVM-LK) provided the best performance on the training dataset (accuracy = 85.2%, sensitivity = 85.1%, specificity = 85.3%, F1-Score = 85.1%). The application of the SVM-LK model on independent datasets confirmed its performances with 88.9% and 80.8% of correct classification for samples collected at D0 and D30, respectively. Conversely, at D10, the proportion of correct classification had fallen to 64.3%. The analysis of saliva samples by MALDI-TOF MS and ML appears as an interesting supplementary tool for COVID-19 diagnosis, despite the mitigated results obtained for convalescent patients (D10).

## 1. Introduction

Since March 2020, the novel coronavirus, severe acute respiratory syndrome coronavirus 2 (SARS-CoV-2), has been considered a pandemic [1]. Its high rate of spread and infection in the human population and the lack of effective and validated treatment has led authorities in several countries to impose lockdowns to slow the spread of coronavirus disease 2019 (COVID-19) [2]. To manage this health crisis, screening individuals is essential to isolate infected cases. These screening tests are mainly carried out using nasopharyngeal swabs (NPSs) and reverse-transcription quantitative polymerase chain reaction (RT-qPCR) for the detection of viral RNA [3]. NPSs are invasive and can cause discomfort from pain to nose bleeding for patients having a sensitive nose [4,5]. This discomfort could induce sneezing by patients, exposing healthcare workers to a significant contamination risk [6,7]. To avoid the constraints of NPS collection, others types of samples were proposed [8,9]. Among them, saliva appeared to be a relevant alternative and had the advantages of being minimally invasive and allowing self-collection [10]. Moreover, we recently demonstrated that the standardization of saliva collection with a Salivette^®^ device improved the molecular detection of SARS-CoV-2 compared to NPSs [11]. Although sensitive and specific, RT-qPCR tests remain relatively long (2–5 h) and expensive, and the strong international demand for molecular biology consumables are factors limiting the implementation of general screening and creating tensions in reagent procurement [12,13].

To prevent the risks of screening shortage, an alternative strategy, PCR-free, based on the detection of specific protein signatures by Matrix-assisted laser desorption ionization time-of-flight mass spectrometry (MALDI-TOF MS) profiling has been proposed. MALDI-TOF MS profiling has been used for more than a decade in routine diagnostics by microbiology laboratories for the identification of microorganisms (e.g., bacteria, yeasts and fungi) [14]. Its performance in terms of sensitivity, specificity, speed, robustness, simplicity of sample preparation and low cost explain the rapid spread of this tool, which has revolutionized identification techniques in microbiology [15,16]. The principle of sample classification relies on the matching of query MS spectra against a library of reference MS spectra. This strategy of spectral matching was successfully applied for viral species identification [17,18].

A pioneering study reported the potential of the MALDI-TOF MS profiling approach to differentiate COVID-19-positive individuals from negative based on the analysis of MS spectra from NPSs [19]. Another work has shown that the combination of MALDI-TOF MS and machine learning (ML) analysis appears to be a suitable strategy for COVID-19 diagnosis using NPS specimens [20]. The application of this strategy to serum samples succeeded in classifying COVID-19 patients according to the severity of clinical symptoms [21]. Among the up-regulated plasma proteins, amyloid A1 and A2 proteins appeared as the best biomarker candidates [21,22]. These data revealed the potential of MALDI-TOF MS profiling for SARS-CoV-2 diagnosis using either blood puncture or NPS samples.

Until now, no work had explored the potential of saliva for the diagnosis of COVID-19 using MALDI-TOF MS profiling. Recently, we established and standardized a protocol for the preparation of saliva samples for MALDI-TOF MS analysis [23]. This work revealed heterogeneity in saliva MS profiles between healthy individuals, which suggested the need for sophisticated bio-statistical analyses, such as ML approaches, to highlight saliva protein signature for COVID-19 diagnosis. In order to address the growing need for screening, the present study investigated the performance of MALDI-TOF MS profiling combined with ML approaches to discriminate COVID-19 patients from healthy individuals applied to saliva MS spectra. Prior to the detection of potential biomarkers, a pipeline of quality control steps from the pre-processing stage was developed to filter out poor quality or non-conforming MS spectra (Figure 1). A set of statistical analyses and five ML approaches were performed on the MS spectra, in order to search for features in spectra distinguishing individuals molecularly diagnosed positive and negative for SARS-CoV-2. Moreover, the ML approach with the best classification performance was applied to a kinetic saliva collection taken from the same individuals ten and thirty days after the primary collection. The corroboration of the classification prediction with the clinical results will be discussed.

## 2. Materials and Methods

### 2.1. Ethical Statement

The study protocol was reviewed and approved by the Ile de France 1 ethical committee (N°2020-A01249-30 protocol, 8 June 2020). Demographics, clinical data, and samples were collected uniquely after the understanding of the study protocol and consent acknowledgement by the participants. A questionnaire on the health status of each participant was carried out. All participant information and samples were anonymized before use. Sample handling was carried out under MSC-AdvantageTM class II biological safety cabinets (Thermo Fischer Scientific, Villebon sur Yvette, France).

### 2.2. Individual Recruitment

COVID group. During the period from 23 July 2020 to 21 September 2020, outpatients consulting at the Institut Hospitalo-Universitaire (IHU) Méditerranée Infection (Marseille, France) and molecularly tested positive for SARS-CoV-2 by NPSs within the last 5 days were invited to enroll in the research study. Saliva samples were collected on the day of patient inclusion (D0), and then ten (D10) and thirty (D30) days later. In parallel, a new NPS was performed for all participants to check their COVID-19 status on D0 and D10. Individuals under 18 years old, non-French-speaking, pregnant women, and individuals suffering of Gougerot-Sjögren Syndrome were excluded. Healthcare workers group. Healthcare workers without fever or respiratory symptoms potentially exposed to patients or patient samples with COVID-19 were invited to enroll in the study. A NPS was performed to all participants to establish their COVID-19 status the same day of saliva collection. Saliva collection was performed on the day of individual inclusion (D0) and then ten (D10) and thirty (D30) days later. Saliva from one healthcare worker (male, 46 years old) was aliquoted (20 µL per tubes), loaded onto each MS plate, and used as inter-plate control.

### 2.3. NPS Management

A standard protocol was used for NPSs collection using nasal swabs with viral transport medium (Pacific Laboratory Products, Blackburn, Australia). Routine diagnosis protocol was applied for SARS-CoV-2 detection on NPS samples by RT-qPCR [24,25]. The classification of individuals was realized according to molecular results of SARS-CoV-2 detection on NPS samples.

### 2.4. Saliva Collection

The optimized protocol for saliva sampling and treatment, previously described, was applied [23]. Briefly, a bottle of spring water was given to each participant who performed a mouthwash prior to saliva sampling. Saliva was collected using Neutral Salivette^®^ (SARSTEDT, Numbrecht, Germany), under the supervision of a laboratory technician. The samples were stored in ice until their use. The storing time never exceeded 6 h. All Salivette^®^ were centrifuged at 1500× *g* for 2 min at 4 °C and were transferred to 1.5 mL Eppendorf tubes and stored at 4 °C until their loading onto MS target plate. All samples were submitted to MS analysis the same day as saliva collection.

### 2.5. Sample Loading

Twenty microliters of saliva sample were mixed with one volume (i.e., 20 µL) of the mixed buffer (mixture (50/50) of 70% (*v/v*) formic acid and 50% (*v/v*) acetonitrile (Sigma-Aldrich, Lyon, France)) used for MALDI-TOF MS profiling analysis [26]. One microliter of the mixed saliva sample was spotted in quadruplicate onto a 96 polished steel MALDI target plate (Bruker Daltonics, Wissembourg, France). After air-drying, 1 µL of matrix solution composed of saturated α-cyano-4-hydroxycinnamic acid (Sigma-Aldrich), 50% (*v/v*) acetonitrile, 2.5% (*v/v*) trifluoroacetic acid, prepared with HPLC-grade water, was added. Matrix quality (i.e., absence of MS peaks due to matrix buffer impurities) and MALDI-TOF apparatus performance were controlled, respectively, by loading matrix solution in duplicate and saliva from the same healthy healthcare worker onto each MS plate. After air-drying at RT, the MS target plate was applied in the mass spectrometer.

### 2.6. MALDI-TOF MS Parameters

Protein mass profiles were obtained using a Microflex LT MALDI-TOF Mass Spectrometer (Bruker Daltonics), with detection in the linear positive-ion mode at a laser frequency of 50 Hz within a mass range of 2–20 kDa. The setting parameters of the MALDI-TOF MS apparatus were identical to those previously used [27]. Briefly, the acceleration voltage was 20 kV, and the extraction delay time was 200 ns. Each spectrum corresponds to ions obtained from 240 laser shots performed in six regions of the same spot and automatically acquired using the AutoXecute of the Flex Control v.3.0 software (Bruker Daltonics).

### 2.7. MS Spectra Preprocessing

Raw data of MALDI-TOF MS spectra (Bruker Daltonics) were preprocessed using MALDIquant [28] and MALDIrppa [29] R packages. The MS spectra were then processed by applying a square root transformation, a smoothing using the Savitzki-Golay-Filter algorithm [30], baseline correction with the SNIP algorithm [31], and normalization of intensity values using the total-ion-current (TIC). All these transformations were performed through the MS processing functions (transformIntensity, smoothIntensity, removeBaseline, calibrateIntensity) of the MALDIquant package using the default parameters. To detect and filter out poor quality (or non-conforming) MS spectra, the screenSpectra function of the MALDIrppa package was used. It computed an atypicality score for each spectrum. All spectra with a score outside the limit were excluded from the rest of the analysis. Then the non-excluded spectra from replicates of the same sample were averaged (averageMassSpectra, MALDIquant package), giving an average spectrum for each sample. The intensity of each average spectrum was normalized by adjusting the TIC (calibrateIntensity, MALDIquant package). The alignment of peaks and spectra were performed with functions of the MALDIquant package. Peaks were detected by adjusting the signal-to-noise ratio (SNR) value and binned with a tolerance value of 0.001. To determine the optimal SNR value, the number of detected peaks and their standard deviation were measured for each SNR value tested. The SNR value for which the number of detected peaks per group was homogenized and stabilized was considered as the optimal trade-off. Only peaks with a frequency of occurrence greater than 50% per group were considered. At the end of the peak detection, the resulting matrix of peak intensities was used for statistical analyses and ML approaches.

### 2.8. Statistical Analyses

After verifying that the values in each group did not assume a Gaussian distribution, Wilcoxon matched-pairs signed-rank tests were computed when appropriate with R. For multiple testing, a false discovery rate (FDR) correction was applied using Benjamini–Hochberg correction [27]. The selection of MS peaks required a significant Wilcoxon rank sum test (*p* < 0.05) and a relevant Benjamini–Hochberg correction (*p* < 0.1).

### 2.9. Dimension Reduction Algorithms

Principal component analysis (PCA) [32] and uniform manifold approximation and projection (UMAP) [33] were used to visualize data on a two-dimensional reduced space and to deduce the relations between spectra profiles of individuals. MS spectra from confirmed SARS-CoV-2-positive patients (*n* = 105) and control healthcare workers (*n* = 51) collected at D0 were used. Dimension reductions were carried out either on all peaks detected or on selected peaks. PCA was performed using the R FactoMineR [34] package. UMAP was computed using the umap R library [35].

### 2.10. Machine Learning (ML) Analyses

MS spectra from 102 saliva samples, including confirmed SARS-CoV-2-positive patients (*n* = 51) and control healthcare workers (*n* = 51) collected at D0, were selected to establish the best algorithm classifying SARS-CoV-2 positives from control samples. Five different algorithms, namely, support vector machine with linear kernel (SVM-LK), support vector machine with radial basis function kernel (SVM-RK), random forest (RF) and k-nearest neighbors (K-NN), and linear discriminant analysis (LDA) were evaluated. ML models were implemented with the caret package [36], except for the SVM-RK, which was performed using the e1071 package [37]. Contrary to caret, the e1071 library provides an implementation of SVM-RK models with the gamma hyperparameter, which allows gamma to be tuned to improve the performance of models. The training process was conducted by a threefold nested repeated tenfold cross-validation to ensure an unbiased estimation of performance. The proportion of the training and test datasets were adjusted to 75% and 25%, respectively, to perform the repeated cross-validation. Model performances after the training step were estimated by computing sensitivity, specificity, accuracy, area under the curve (AUC), and average precision (AP) values. AUC and AP were computed on receiver operating characteristic (ROC) and precision-recall (PR) curves with the ROC R package [38].

## 3. Results

### 3.1. Characteristics of Participants

A total of 360 saliva samples, obtained from 138 patients diagnosed SARS-CoV-2 positive (Cov+ group) and 51 healthcare workers considered as healthy control (Cov− group), were included in the study. For both groups, serial saliva sampling was performed ten (D10) and thirty (D30) days after the first saliva collection (Table 1). Among patients from the Cov+ group, the SARS-CoV-2-positive status was confirmed by NPS tests for 76.1% (105/138) of them at the enrollment day (D0). At D10, the virus was still detectable for 16 out of 79 (20.3%) patients from the Cov+ group. At D30, the 20 individuals from the Cov+ group were considered as exempt of infections based on the delay since primary infection and in the absence of any clinical symptoms in the ten days before saliva sampling. SARS-CoV-2 detection performed on NPS samples from healthcare workers (Cov− group) were all negative at the enrollment day (D0, *n* = 51) and ten days later (D10, *n* = 40) (Table 1). The absence of any clinical symptoms in the ten days before saliva sampling at D30 for the 32 individuals from the Cov− group were considered as exempt from infections. Moreover, SARS-CoV-2 was detected in none of the individuals from Cov+ and Cov− groups molecularly tested on saliva at D30. No significant differences were noted for age (*p =* 0.069, Kruskal–Wallis test) or gender (*p = 0.279*, df = 5, Pearson’s Chi-square test) among the groups considering collection time points. It is interesting to note that all individuals from the Cov+ group were tested positively, outside our institute, less than one week before in order to be included in the study.

### 3.2. Preprocessing Steps of Saliva MS Spectra

Prior to researching the saliva protein signature distinguishing Cov+ from Cov− individuals, MS spectra were submitted to successive quality control steps. As each sample was loaded in quadruplicate, a total of 1440 MS spectra were obtained. The screenSpectra function (MALDIrppa package) assigned an atypicality score (score threshold > 0.9, Figure 2A) to 23 MS spectra (Cov+, *n* = 18; Cov−, *n* = 5). These spectra were rejected for the rest of the analysis. As no more than two replicate spectra were excluded for each sample, no saliva sample was removed after this step. After filtering atypical spectra and averaging spectra replicates, the comparison of the TIC between Cov+ (*n* = 237) and Cov− (*n* = 123) groups revealed no significant differences (*p* > 0.05, Wilcoxon rank sum test, Figure 2B), underlining a stable spectra acquisition throughout the study independently of the sample group origin.

The comparison of peak numbers detected according to the SNR value was performed to select the SNR value allowing for the inclusion of mainly true peaks and assessing peak detection between the Cov+ and Cov− groups. For SNR values lower than 4, the number of detected peaks was highly heterogeneous between the two groups and among the SNR values. Moreover, an important decrease in the number of detected peaks was noticed with the increment of SNR values (Figure 2C). Conversely, for SNR values of 4 and above, the mean number of detected peaks and its standard deviation diminished slowly (Figure 2D). Interestingly, the peak number distributions between the Cov+ and Cov− groups were not significantly different only for the SNR value equal to 4 (*p* > 0.05, Wilcoxon rank sum test, Figure 2D). Then, for the remainder of the analysis, a SNR value of 4 was selected and an intensity matrix of 212 peaks was obtained.

### 3.3. Comparison of Saliva MS Spectra and Detection of Potential Biomarkers

For these analyses, unique saliva samples from Cov+ group confirmed SARS-CoV-2 positive at D0 (*n* = 105) were compared to the Cov− group from D0 (*n* = 51). No visible difference was noticed when the mean spectra of the Cov+ (*n* = 105, Figure 3A) and the Cov− (*n* = 51, Figure 3B) groups were compared (Figure 3C), reflecting the complexity and the heterogeneity of spectra profiles. The application of a Wilcoxon rank sum test (*p* < 0.05) identified 52 peaks distinguishing SARS-CoV-2 positive from negative samples (Appendix A). The correction for multiple hypothesis testing (Benjamini-Hochberg, *p* < 0.1) allowed nine peaks to be selected (Figure 3D). Among these peaks, seven and two exhibited higher intensity in SARS-CoV-2-positive and negative groups, respectively (Figure 3E). The peaks with mass-to-charge ratio (m/z) of 2489.2 and m/z of 5418.9 possessed the most significant intensity differences for the positive (Figure 3F) and negative (Figure 3G) groups, respectively. PCA analysis, carried out on these 9 selected peaks, allows a partial distinction between SARS-CoV-2-positive and negative groups (Figure 3H). Carrying out a UMAP on these selected peaks does not improve the separation of groups (Figure 3I). When PCA and UMAP were performed using all peaks (Appendix A) or the 52 peaks selected with two-tailed Wilcoxon rank sum test (Appendix A), no clear separation could be noticed.

### 3.4. Building and Selection of Machine Learning Models

To assess the performance of ML approaches to separate SARS-CoV-2-positive from negative individuals, samples were split into different groups. A first group was used during the building and training phase of the ML models, which allowed the selection of the best model. Then, during the test phase, the performance of the selected model was evaluated on independent groups. For building the training model, a balanced group of samples (51 Cov+, 51 Cov−) was formed from individuals at D0 (Figure 4). The 51 Cov+ samples were randomly selected from the 105 positive samples at D0 with a confirmed infection status. The remaining samples were organized into five test groups (Test 1 to Test 5), according to collection time points (D0, D10, and D30) and NPS RT-qPCR results (Figure 4). A total of 96 samples from Cov+ group were classified as uncertain negative for SARS-CoV-2, because all these individuals were diagnosed as SARS-CoV-2 positive either in the week preceding their enrollment (*n* = 33 at D0) or ten days before this second NPS test (*n* = 66 at D10). Considering a SARS-CoV-2 false-negative, the detection rate can reach 30% [39], these 96 samples were classified as uncertain negatives and were tested separately (Test 5).

Five different ML models (K-NN, LDA, RF, SVM-LK and SVM-RK) were studied. The best ROC and PR curves obtained for each ML model are presented in Figure 5A and Figure 5B, respectively. With the exception of the K-NN model, the models exhibited high performance with the best area under the curve (AUC) values greater than 0.92 and 0.94 for ROC and PR curves, respectively. To select the model with the best compromise of sensitivity and specificity, the F1-score was computed (Figure 5C). The SVM-LK model obtained the best F1-score, about 0.85, achieving a sensibility of 85.1% and a specificity of 85.3%. The SVM-LK model reached the greatest average area under the curve (AUC) values of 0.93 for ROC and 93 for PR curves, supporting the existence of a protein pattern associated with saliva mass spectra. The SVM-LK model was then applied to test groups to determine its prediction performance.

### 3.5. Performance of the SVM-LK Model

To evaluate the performance of the SVM-LK model, all saliva MS spectra were included except those used for model training. The first test set (Test 1, *n* = 162), included MS spectra collected at D0, D10, and D30 from Cov+ (*n* = 90) and Cov− (*n* = 72) groups, with the exception of samples classified as uncertain negatives. Saliva samples collected at D30 from both groups (Cov+ and Cov−) were classified as SARS-CoV-2 negatives (*n* = 52). Then, among Test 1, 70, and 92, samples were classified as positive and negative for SARS-CoV-2, respectively, based on NPS RT-qPCR results (Figure 4). On the Test 1 set, the SVM-LK model distinguished SARS-CoV-2-positive and negative samples with accuracy, sensibility, and specificity of 77.8%, 78.6%, and 77.2%, respectively (Figure 6A,E). The model performance on this independent dataset was lower than those obtained during model training. Interestingly, among the 70 individuals classified as SARS-CoV-2 positives by NPSs from Test 1, 15 saliva samples were not confirmed by the SVM-LK model (Figure 6A). The comparison of viral loads revealed that the Ct values from the 55 samples (mean ± SD, range: 24.6 ± 5.8, 10.0–33.8) classified positives by both approaches, were significantly lower (*p* < 0.02, Mann–Whitney test) than those from 15 saliva samples (mean ± SD, range: 28.2 ± 6.2, 10.0–34.3) classified negatives by ML. These results support the hypothesis that samples with higher viral loads were better classified by SVM-LK model.

The model, trained only on samples collected at D0, could be less efficient in classifying samples collected at D10 and D30. To assess this hypothesis, the SVM-LK model was applied to MS spectra from samples collected at D0 (Test 2, *n* = 54), D10 (Test 3, *n* = 56), and D30 (Test 4, *n* = 52) separately (Figure 6B, Figure 6C and Figure 6D). The proportions of correct classification were of 88.9% for Test 2, 64.3% for Test 3, and 80.8% for Test 4 (Figure 6E). Interestingly, higher performance was obtained for Test 2, which includes samples collected at D0 as those used to train the model. Finally, we applied the model to MS spectra from samples classified as uncertain negative samples (Test 5, *n* = 96). More than half of the samples (*n* = 54/96) remained predicted as SARS-CoV-2-positive samples (Figure 6F): two third (*n* = 22/33) at D0 and half (*n* = 32/63) at D10.

## 4. Discussion

To overcome the constraint of NPS sampling and the limitations of molecular diagnosis of COVID-19, the use of saliva specimens combined with the MALDI-TOF MS and ML approaches appeared as a promising alternative strategy. The present work described the application of MALDI-TOF MS for the identification of a specific salivary protein pattern to distinguish SARS-CoV-2-positive from negative individuals. For this purpose, the entire analytical chain must be of high quality.

The quality of spectra is directly linked to the mode of sample collection, storing, and preparation before submission to MS [40,41]. Although passive drooling is the most simple and common method for saliva collection, the use of specific devices, such as Salivette^®^, allows the specimen collection to be standardized [11,42]. Moreover, the saliva is composed of numerous enzymatic proteins that could alter MS spectra by auto-digestion of the sample [43]. To limit these phenomena, an optimized protocol for saliva collection and management applied for MS analyses has recently been established and was used in the present work [23]. Although this protocol enables excellent intra-individual reproducibility, the salivary profile between individuals is heterogeneous. This heterogeneity requires the acquisition of high-quality MS spectra to succeed in a correct classification. In this way, pre-processing steps evaluating MS spectra quality and their homogeneity between groups are crucial. They were carefully defined and performed in this work.

Filtering of atypical MS spectra with MALDIrppa package was performed on each replicate. The rejection of spectra was operated before averaging spectra per sample to prevent alteration of protein profiles. No significant difference in TIC distribution between the Cov+ and Cov− groups was determined, which ensured homogeneity of the spectra intensities. Quality control steps excluding outlier MS spectra improved the comparison of MS spectra. Although the filtering step is frequently used for the pre-processing of MS spectra [19], the TIC distribution is seldom considered. Despite TIC results being linked to the sample composition, important variations of TIC could occur between samples of the same type. These TIC variations generally reflect experimental or technical variations such as sample preparation (e.g., treatment or amount) and apparatus performances (e.g., laser power or detector sensitivity) [44,45], which could hamper the detection of pertinent biological information. The application of a normalization to TIC values allows all samples to be equalized.

The establishment of peak detection threshold is another decisive parameter for the detection of biological markers [46]. Peak detection depends on the determination of the optimal SNR value. Applying an insufficiently strict SNR value increases the risk of including peaks corresponding to background noise, which could distort the statistical analysis and lead to an inappropriate interpretation of the results. Inversely, choosing too high an SNR value may lead to the miss-detection of true peaks, and this lack of information could be deleterious for further analysis steps. Here, the analysis of the number of peaks detected and its variation among samples according to the SNR value, allowed the selection of a threshold of 4. Indeed, for SNR values lower than 4, the high number of peaks detected and its high dispersion among samples were likely related to peaks detected in the background of spectra. Moreover, at an SNR of 4, we controlled that the number of peaks detected between the Cov+ and Cov− groups were comparable, limiting any bias in the analysis. The recent studies evaluating the performance of MALDI-TOF MS to discriminate SARS-CoV-2-positive from negative individuals, using either NPS or serum samples, did not clearly detail the criteria for selecting the SNR value [19,21,47].

For serum samples, the selection of an SNR value of 5 appeared sufficient to separate COVID-19 patients into three levels of severity by applying ML models [47]. Nevertheless, the stringency of peak detection might miss some specific peaks that could improve this classification. Conversely, Nachtigall et al. chose an SNR value of 2 for peak detection of NPS from individuals diagnosed as positive or not for COVID-19 [19]. Based on this SNR value, they determined seven significantly different peaks. These discriminant peaks had a very low intensity and could likely not be detected if a higher SNR value was selected. Generally, the SNR value is chosen to be high enough to prevent the detection of peaks in background noise. However, it is likely that too high an SNR value will filter out decisive peaks for classification. Therefore, establishing an accurate strategy to optimize the SNR value becomes compulsory to avoid miss- or over-detection of peaks. The application of these quality control steps should reduce the number of replicates from four to two per sample, allowing 48 samples to be tested in less than 2 h [14].

Using an SNR value of 4 at the peak detection step, we obtained an intensity matrix of 212 peaks. Among them, nine peaks with a relevant different intensity distribution were identified between positive samples for SARS-CoV-2 and healthy individuals. A significantly higher intensity distribution for the Cov+ group compared to the Cov− group was observed for 7 of the 9 peaks. Natchigall et al. [19] reported in nasal swab samples that five of the seven peaks, distinguishing Cov+ and Cov− groups, have a higher intensity distribution in healthy individuals. Similarly, in another study using swab samples, all of the discriminant peaks possessed a higher intensity distribution for uninfected individuals than for SARS-CoV-2-contaminated patients [20]. Conversely, here in saliva samples, discriminant MS peaks correspond mainly to up-regulation protein expression subsequent to SARS-CoV-2 infection.

In parallel to the differential analysis of peak intensity, none of the dimension reduction algorithms (i.e., PCA or UMAP) applied to the intensity matrix of the nine selected peaks were able to show a clear separation between SARS-CoV-2-infected and healthy individuals. Similar observations were noticed by other studies examining MS spectra from Covid-19 patients [19,22]. Effectively, only partial distinctions could be identified between infected and uninfected individuals. Based on the above observations, we can assume that dimension reduction algorithms struggle to separate groups because the inter-individual heterogeneity of spectra profiles is large and stronger than the differences between infected and non-infected groups. These intergroup differences are due to a few biomarkers the differential expression of which contributes little to the overall heterogeneity.

As the differences between the groups are masked by the overall heterogeneity, five ML models were implemented to specifically identify and combine protein profiles in order to increase their effectiveness in separating the groups. The training process of the models allowed us to determine that the SVM model with linear kernel was the best model, based on the F1-score [12,47]. Saliva spectra were classified with a high accuracy (i.e., 85%). These results are comparable to or better than the performance of ML models already described for COVID-19 diagnosis [12,20]. Several studies established that the SVM model performed the best predictions [12,19,21].

To confirm the performance and the robustness of the selected classifier, the SVM-LK model was applied to independent datasets (Test 1 to 5). First, an independent set was constructed by including all saliva spectra, excluding the spectra used in the training set and those classified as uncertain negatives. The performance on this set was a little lower than on the training set, with an accuracy of 77.8%. The spectra were then split according to the collection time (i.e., D0, D10, and D30). The separate evaluation on these independent datasets revealed good predictions at D0 (89%) and D30 (81%). In contrast, at D10, a low accuracy was obtained, notably, due to the incorrect classification of SARS-CoV-2-positive samples by the ML model (sensitivity lower than 45%). The weak performance of the ML model on this test set may be due to the gradual recovery of infected patients. It could be hypothesized that a progressive disappearance of the protein pattern linked to the spectra classification in the Cov+ group occurred during the recovery situation. A recent study demonstrated that the concordance of correct classification between saliva and NPS samples by RT-qPCR was lower for convalescent patients [11], and which was corroborated by others [48]. The mitigated results of classification were attributed to an intermediate status of patients, whom are no longer at peak disease but not yet fully recovered.

Test 5 consists of patients defined as uncertain negative samples for SARS-CoV-2. The discrepancy in classification between the molecular results and the model prediction can be explained by several factors. First, as all individuals from the Cov+ group were diagnosed molecularly as positive for SARS-CoV-2 a few days before enrollment, it is likely that several of them remained positive on the enrollment day (D0) and that the incorrect prediction could be attributed to a molecular miss-detection of the virus on NPSs. Effectively, as the initial saliva sample (D0) was carried out less than 5 days after the onset of symptoms (median days—IQR: 2.2—1–3), it was expected that their COVID-19 infectious status was molecularly confirmed at the enrolment day (D0) for nearly all of them. Unexpectedly, SARS-CoV-2-positive RT-qPCR were confirmed for solely 76.1% (*n* = 105/138) of the patients. Previous studies reported that the risk of false-negative diagnosis is not infrequent and was estimated around 30% by molecular detection of the virus in NPSs [49]. Furthermore, others reported that COVID-19 miss-diagnosis could reach up to 50% of suspected cases using NPS specimens [39,50]. For these reasons, patients from the Cov+ group, classified negatives by RT-PCR on NPSs at D0 and D10, were considered as uncertain negatives to avoid miss-interpretation and learning errors. Retesting of NPS samples to control the rate of false-negatives would be advisable, nevertheless, it was not possible because the molecular detection of SARS-CoV-2 on NPS samples was performed in the frame of routine laboratory diagnosis and we do not have access to these samples. Despite a negative RT-qPCR result, the persistence of virus molecules or infection-related proteins would likely occur, and sustainability of serological inflammatory markers several days or weeks after patients recovery was reported [51,52]. These long-term protein dysregulations could perturb ML classification. 

The second hypothesis could be that the virus was recently cleared from these individuals, at least in the nasopharyngeal part, and that they are therefore truly negative for SARS-CoV-2. Nevertheless, despite the disappearance of the virus, it is conceivable that traces of the infection could persist for a longer period of time leading to incongruent results between the ML model and the molecular analysis. The correct classification of more than 80% of the samples at D30 by the SVM-LK model comforts this claim. We noted that the majority of studies did not evaluate performances of ML models on an independent dataset [19,21], but established the performance metrics on the training set. Two studies tested the ML models on an independent dataset [12,20]. The performance of the ML models was noticeably lower than that obtained on the training set, suggesting an overfitting of the model to the training data. To avoid this problem and to obtain an unbiased measure of the performance of the model, it is essential to use an independent test set [12]. Interestingly, the exclusion of samples for which doubt persisted after molecular classification (i.e., convalescent patients from Test 3 and uncertain negative samples from Test 5) allowed us to establish the performances of the SVM-LK model. Pooling samples from Test 2 (patients collected at D0 with molecular confirmation of SARS-CoV-2 infection) and Test 4 (recovered patients collected at D30 and healthy individuals) revealed a sensitivity, specificity, and accuracy of 88.9%, 80.8%, and 84.9%, respectively.

The main originality of our study was the use of saliva as the specimen. Although previous works reported the potential of MALDI-TOF MS to monitor SARS-CoV-2 proteins and host immune response against the virus using water gargle samples as saliva collection [53,54], these specimen collections would be unappropriated for young children or the elderly. Unlike NPSs, saliva collection is a non-invasive technique which can be performed by the patients themselves and prevents the risk of contamination for healthcare workers [55,56]. Previous works reported the potential of MALDI-TOF MS to monitor SARS-CoV-2 proteins and host immune response against the virus using water gargle samples as saliva collection [53,54]. The introduction of the Salivette^®^ device has standardized the collection and improved the molecular detection of SARS-CoV-2 compared to NPSs [11]. However, it has recently been reported that mouth washing before collection impaired the detection of the virus in saliva [57], that is why here the molecular results of NPSs were used as a reference for sample classification. A mouth washing was systematically proposed to all individuals enrolled in the present study prior to saliva collection, as numerous patients ate and/or drank while waiting for their turn at the hospital. The record attendance occurred during the second wave of COVID-19 outbreak in the south of France, resulting in a long wait to be diagnosed. It is possible that the mouth washing could also have reduced the protein signatures associated with SARS-CoV-2 infections and thus altered the performance of ML models for sample classification. To assess the effect of mouthwash on model performance, complementary experiments are required in which individuals who ingested drink or food in the hour prior salivation collection will be excluded [58]. 

Recently, a work associating RT-PCR and MALDI-TOF MS was applied for SARS-CoV-2 detection in saliva samples [59]. Conversely to the present study which researched a specific protein signature associated to SARS-CoV-2 infection, Hernandez et al. targeted, by MALDI-TOF MS, the detection of RT-PCR products from the virus [59]. The main advantages of the RT-PCR/MALDI-TOF MS-based system are a direct detection of amplicons from the virus and the possibility to detect simultaneously several targets. Although, the saliva quantity (about 300 µL) and time required to obtain the results (> 8 h) are more important than the strategy used here, the opportunity to realize multiplexing could allow virus mutations and consequently SARS-CoV-2 variants to be identified.

The performance of ML models to predict clinical symptoms in SARS-CoV-2 positive patients will also be evaluated using saliva as the specimen. The characterization of such protein markers will be used to predict the favorable or unfavorable clinical course of a patient in order to guide physicians on the actions to be taken. A recent study reported that the application of ML models on serum samples succeeded in predicting the clinical evolution of SARS-CoV-2-positive patients [21]. The main limitation of the present work was the lack of saliva samples from individuals infected by another virus such as influenza. The enrollment of individuals with flu symptoms was initially planned in order to test whether it was also possible to separate COVID-19 infections from other respiratory viruses. Unfortunately for our study, during the 2020 summer period in France, the application of barrier gestures and social distancing participated likely in the prevention of the emergence of respiratory diseases and therefore it was impossible to recruit patients from this group. The potential of ML models to distinguish SARS-CoV-2 from other respiratory viruses, including influenza, has already been demonstrated based on the analysis of MS profiles from NPS [20].

The aim of works using MALDI-TOF MS profiling plus ML models for COVID-19 diagnosis was not to obtain a direct detection of the viral proteins, but rather to identify a specific protein signature associated to viral infection [19,20]. Characterization of the protein(s) involved in this signature is not required at all for successful classification. Characterization of the signature peaks can be carried out in a second step to determine the identity and origin of the proteins involved. Two recent studies [21,22] applying a similar strategy onto NPS or serum specimens demonstrated that the identity of proteins beside MS peaks that allowed for the classification of the samples was of human origin (i.e., serum amyloid proteins A1 and A2), confirming that the detection of viral proteins were not necessary for correct classification of the sample.

A recent publication [60] established an original COVID-19 diagnostic strategy to detect selected viral peptides in NPS specimens. It combines immuno-affinity purification followed by high resolution MS-based target assay and parallel monitoring reaction assays (PRM). The application of ML models on data from PRM assays allowed a sensitivity of 97.8% and specificity of 100% to be reached compared to qRT-PCR for determining COVID-19-positive samples. Moreover, the performance of this strategy compared to Ag RDTs (rapid antigen diagnostic tests) revealed a significantly higher sensitivity for the MS antigen approach compared to nearly all Ag RDTs assessed (3 among 4 tested) [61]. These works confirmed the high potential of MS approaches for the clinical diagnosis of infectious agents, which could be helpful when tensions occurred in molecular reagent supply or when the systems are saturated.

In conclusion, MALDI-TOF MS with the collection of salivary samples offers real new perspectives of SARS-CoV-2 clinical diagnostics. The non-necessity of RT-qPCR test and the affordable cost of the MALDI-TOF MS analysis per sample could allow a massive test of the population in order to distinguish infected from uninfected individuals, thus bringing potential improvements to sanitary conditions.

## Figures and Tables

**Figure 1 jcm-11-00295-f001:**
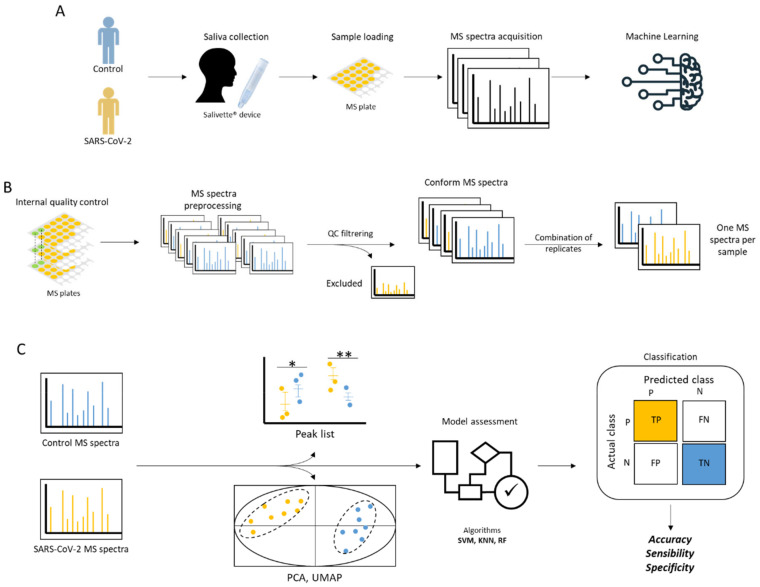
Schematic presentation of the experimental workflow. (**A**) Main steps of the study. Individuals who were SARS-CoV-2-positive and negative (controls) based on NPS RT-qPCR tests were enrolled. Saliva was collected with Salivette^®^ devices (SARSTEDT, Numbrecht, Germany), loaded onto an MS plate, and then submitted to MALDI-TOF MS acquisition. The resulting MS spectra were analyzed using machine learning (ML) methods. (**B**) Pre-processing steps of the MS spectra included a quality control step, evaluated the homogeneity of the data among the Cov+ and Cov− groups, and determined peak detection conditions. (**C**) Strategy used for the prediction of SARS-CoV-2-positive saliva using ML. Statistical analyses were performed to reveal relevant MS peaks before the assessment of ML models.

**Figure 2 jcm-11-00295-f002:**
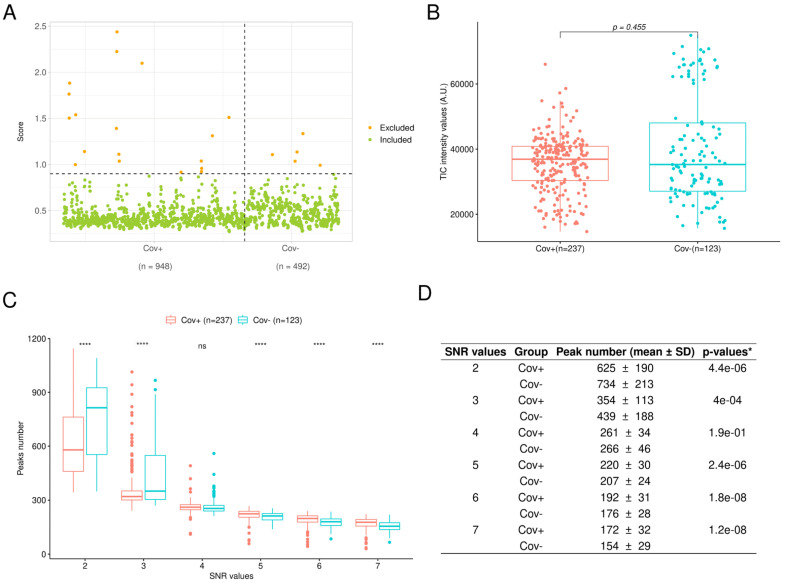
Pre-processing steps of MS spectra. (**A**) Detection of atypical spectra from Cov+ (*n* = 948) and Cov− (*n* = 564) groups using the screenSpectra function from the MALDIrppa package. A score was computed for each spectrum. Horizontal dotted line corresponds to atypical score threshold. (**B**) Comparison of total-ion-current intensities after filtering atypical spectra and averaging spectra replicates between Cov+ (*n* = 237) and Cov− (*n* = 123) groups (two-tailed Wilcoxon rank sum test). (**C**) Comparison of peak number distributions according to SNR values ranging from 2 to 7 between average MS spectra from Cov+ (*n* = 237) and Cov− (*n* = 123) groups. **** *p* < 0.001 by two-tailed Wilcoxon rank sum test. (**D**) Summary table of peak number distributions and comparisons for each SNR value between average MS spectra from Cov+ (*n* = 237) and Cov− (*n* = 123) groups. * exact *p-*values of by two-tailed Wilcoxon rank sum test. A.U.: arbitrary units; Cov+/−: MS spectra from individuals enrolled in the COVID-19 and control groups, respectively; SD: standard deviation; SNR: signal-to-noise ratio.

**Figure 3 jcm-11-00295-f003:**
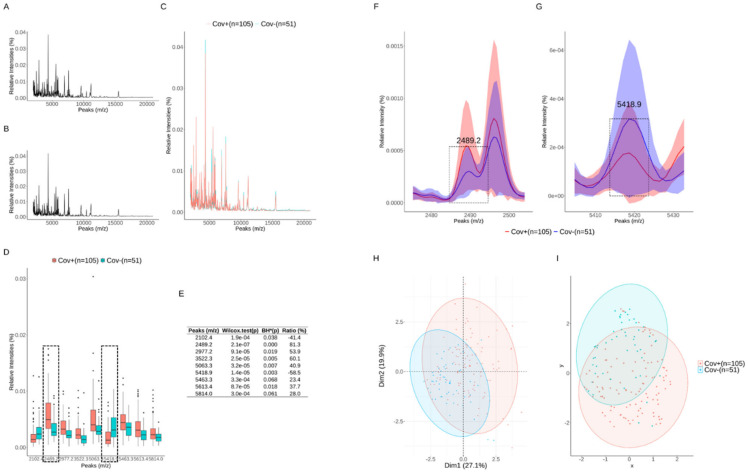
Comparison of saliva MS spectra between SARS-CoV-2(+) (*n* = 105) and control (*n* = 51) groups sampled at D0. Mean MS spectra profiles with a total-ion-current normalization applied to peak spectra intensities of Cov+ (**A**) and Cov− (**B**) individuals and their superposition (**C**). (**D**) List of MS peaks with significant total-ion-current normalized intensity differences (two-tailed Wilcoxon rank sum test (*p* < 0.05) with a Benjamini–Hochberg correction (*p* < 0.1) for each peak) between Cov+ and Cov− groups. Dashed box highlighted the most two relevant peaks. (**E**) Summary table of Wilcoxon rank sum tests with a Benjamini–Hochberg (BH) correction for each peak. Ratio represent the proportion (%) of mean peak intensity variations between Cov+ and Cov− groups. Positive and negative values correspond, respectively, to upper and lower mean peak intensity variations in Cov+ compared to Cov− groups. Comparison of the two greatest peaks at m/z 2489.2 (**F**) and at m/z 5418.9 (**G**) differentiating Cov+ and Cov− groups. Lines and shadow represent mean values ± interquartile range (IQR), respectively. Dashed box framed each relevant peak. PCA (**H**) and UMAP (**I**) performed on Cov+ (*n* = 105) and Cov− (*n* = 51) groups using the selected peaks (*n* = 9) with two-tailed Wilcoxon rank sum test (*p* < 0.05) and a Benjamini–Hochberg correction (*p* < 0.1).

**Figure 4 jcm-11-00295-f004:**
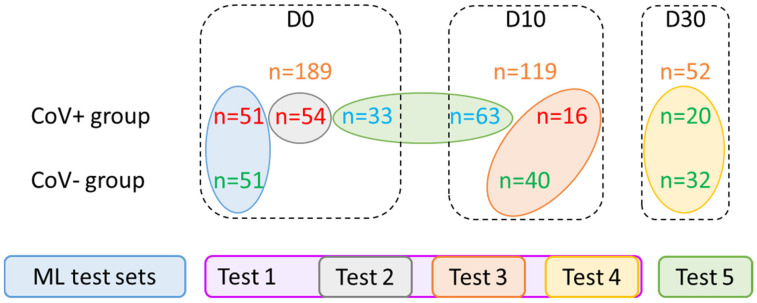
Training and testing sets for machine learning models. The individuals enrolled were divided into several groups to establish the best training set and to assess the performance of the selected ML model on different groups. The total number of samples, in the Cov+ (*n* = 237) and Cov− (*n* = 123) categories, at the inclusion (D0), ten (D10), and thirty (D30) days later were indicated (orange numbers). Green, red, and blue numbers correspond to saliva samples classified as SARS-CoV-2 positive, negative, or uncertain negative, respectively, based on RT-PCR results. The samples selected for training set selection (*n* = 102) and assessment of the best ML model, organized into five groups (Test 1 to Test 5) are indicated by circles of the same color code. Test 2 to Test 4 were included in Test 1. Cov+/−: MS spectra from individuals enrolled in the COVID-19 and control groups, respectively; ML: machine learning.

**Figure 5 jcm-11-00295-f005:**
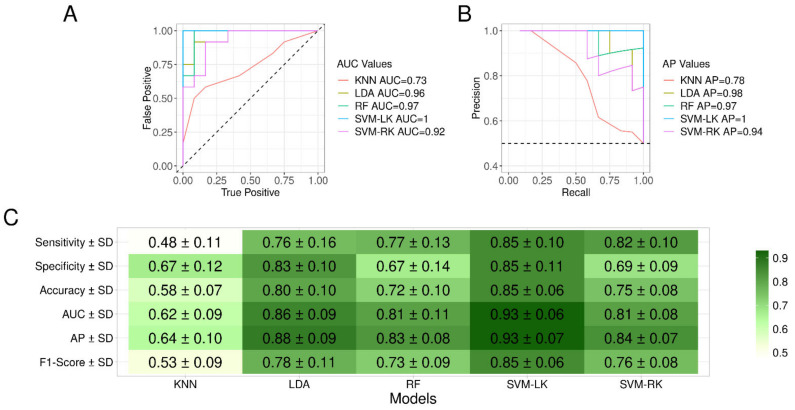
Performances curves of the five machine learning models. Best AUC values for ROC (**A**) and best AP values for precision recall (**B**) curves associated to each model trained on samples from Cov+ (*n* = 51) and Cov− (*n* = 51) groups collected at D0 are presented. (**C**) Summary of performances obtained per ML model. F1-Score was used as a performance measure to select the most suitable developed model. AP: average precision; AUC: area under the curve; Cov+/−: MS spectra from individuals enrolled in the COVID-19 and control groups, respectively; K-NN: K-nearest neighbors; LDA: linear discriminant analysis; ML: machine learning; RF: random forest; SVM-LK: support vector machine with linear kernel; SVM-RK: support vector machine with radial kernel.

**Figure 6 jcm-11-00295-f006:**
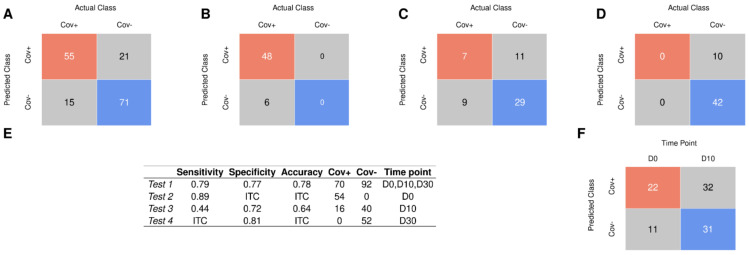
Confusion matrix for test results using SVM linear model. The metrics of the prediction using samples from the Test 1 (**A**), Test 2 (**B**), Test 3 (**C**), and Test 4 (**D**). (**E**) Summary of performances obtained per group tested (from Test 1 to 4) applied to the SVM-LK model. ITC: impossible to compute. (**F**) The metrics of the prediction using samples from Test 5 applied to the SVM-LK model. SVM-LK: support vector machine with linear kernel.

**Table 1 jcm-11-00295-t001:** Characteristics of participants.

	COVID-19 Group ^a^			Healthcare Worker Group		
Collection Time Point ^b^	D0	D0 + 10	D0 + 30	D0	D0 + 10	D0 + 30
Participants, *n* (COVID+ ^c^, *n*)	138 (105)	79 (16)	20 (0)	51	40	32
Age (years), median (IQR)	37.4 (23–52)	37.7 (24–52)	48 (39.5–57.3)	36.1 (27–45.5)	38.4 (28.8–46.8)	37.7 (27.8–46.8)
Male, *n* (%)	68 (49.3%)	42 (53.2%)	10 (50.0%)	22 (43.1%)	18 (45.0%)	16 (48.5%)
Onset of symptoms before D0 test (days), median (IQR)	2.2 (1–3)			/		
Symptoms at presentation, *n* (%)	89 (64.5%)			0 (0.0%)		
Headache, *n* (%)	38 (27.5%)			/		
Tiredness, *n* (%)	26 (18.8%)			/		
Cough, *n* (%)	24 (17.4%)			/		
Fever, *n* (%)	21 (15.2%)			/		
Myalgia, *n* (%)	20 (14.5%)			/		
Breathing difficulties, *n* (%)	12 (8.7%)			/		
Anosmia/Ageusia, *n* (%)	9 (6.5%)			/		
Sore throat, *n* (%)	7 (5.1%)			/		
Diarrhea, *n* (%)	7 (5.1%)			/		
Others, *n* (%)	4 (2.9%)			/		

^a^ Tested positively for SARS-CoV-2 by RT-qPCR on NPSs less than five days before enrollment. ^b^ Saliva sampled ten (D0 + 10) and thirty (D0 + 30) days after the first collection (D0). ^c^ Tested positively for SARS-CoV-2 by RT-qPCR on NPSs the sampling day (D0). Abbreviations: IQR, interquartile range; NPS, nasopharyngeal swab; SARS-CoV-2, severe acute respiratory syndrome coronavirus 2.

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
