# Peer review of "Exploratory Study on Application of MALDI-TOF-MS to Detect SARS-CoV-2 Infection in Human Saliva"

_jcm, 2022, doi:10.3390/jcm11020295_

Round 1

Reviewer 1 Report

The authors have developed a MALDI-TOF-MS method to detect SARS-CoV-2 from COVID-19 positive individuals. They have performed extensive work and have a track record of developing MALDI-ToF based methods, but for the acceptance of this article, they must address the following issues-

  1. The authors have claimed that their work is the first one using Saliva to detect SARS-CoV-2 using MALDI-ToF. Although they have used Neutral Salivette here, Hernandez et al. published a similar work recently. Authors are requested to mention the differences and advantages in their manuscript.
  2. The authors are requested to explain why they exclude individuals under 18 from the study and non-french speaking ones.
  3. The authors are requested to compare the MALDI-ToF results with CT values to better explain for not receiving a positive result with MALDI-ToF when the gold-standard RT-PCR was positive.
  4. The authors took RT-PCR negative samples as negative standard but ignored the false-negative results here, whereas they are claiming false-positive results in cases of mismatches. The authors are requested to rerun the samples to clarify the issue.
  5. To confirm the presence of SARS-CoV-2, authors should spike the negative Saliva with inactivated viruses. 
  6. The authors have developed the method without cross-reacting samples, although they mention the limitation briefly, but how do authors justify the detection of COVID-19 infection with their method? An infected individual with another respiratory virus/pathogen may show closely related peptide mass fingerprints. 

Reviewer 2 Report

COVID-19 diagnosis is critical in diagnosis and treatment of SARS-CoV-2. RT-qPCR from NPS samples is currently the gold standard reference test, but it has drawbacks such as cost and time to results. Saliva is another avenue to diagnose SARS-CoV-2 via a PCR-free method. In particular, MALDI-TOF MS combined with ML is up 80-89% effective in classifying patients as compared with RT-qPCR on D0 and D30, but closer to 65% on D10. Thereby, providing an alternative or supplemental approach for testing.

Comments:

  • Do make sure that there are sufficient references to make sure the text is not riddled with anecdotes, such as lines 56-58. It makes sense that NPS is invasive that is discomforting causing sneezing thereby increasing contamination to front-line workers. However, a reference or two would make the argument stronger. Otherwise, if it is not a strong point it is not required and can be omitted since it detracts from the main thread.
  • For the argument about reagent procurement and screening shortage for PCR based tests in 67-77. Do you have a reference or rough calculation based on comparable price or availability can MALDI-TOF MS profiling cover the same capacity as current RT-qPCR tests? or if not 1-1, how much strain could it relieve? 
  • Could you provide a comparable time comparison for RT-qPCR and MALDI-TOF MS? The point on line 64 is that PCR is 2-5 hours, which is stated to be relatively long, but no comparison on MS-ML timing. How long does it take to analyze a single patient result? How many duplicates are required? Are 4x MALDI-TOF MS replicates necessary/standard in the field? 
  • Given the dataset and sample size was there any information on robustness? Meaning is there greater sensitivity for particular variants of SARS-CoV2?
  • Was there a criterion for use of the 5 ML models in the study? Are there references for use of particular models in the MS field? Are there more optimized models for different types of MS data?
  • Since positive (or negative) results heavily rely on the ML analysis on the MS data, would there be a need for more training or re-training before broader use? How universal is the model?
  • What computation was required for the ML analysis? training?
  • Are there particular differences between D0/D30 and D10 to account for the 20% change in the test accuracy?
  • In line 540 there was a discussion on mouthwash, or food/drink intake. Is there any evidence to show that the protein makeup significantly changes due to these parameters? Are there other factors that may impact the analysis as greatly as the D0 vs D10 vs D30 differences?

Round 2

Reviewer 1 Report

The authors have responded to most of the quarries but are requested to briefly add the limitations (previous quarries 4, 5, 6) in the manuscript. 

Author Response

This manuscript is a resubmission of an earlier submission. The following is a list of the peer review reports and author responses from that submission.

Round 1

Reviewer 1 Report

The manuscript by Costa et al. describes the detection of SARS CoV-2 for the differentiate COVID-19 positive patients from COVID-19 negative patients. Authors have analyzed saliva samples from COVID-patients at different time points. MALDI-TOF MS analysis followed by data analysis using machine learning approach was used to distinguish COVID-19 positive patients from the negative patients with reasonably good specificity and sensitivity. Non-invasive sample such as saliva has been used to identify signature spectra which further used to develop the algorithm to differentiate diseased and healthy patients.

The present work is of interest to community but I do see some issues with the manuscript which I will describe below.

  1. As authors have described, saliva samples for this study were collected on D01, D10 and D30 and were stored at 4°C until further analysis. Have author performed the stability of the collected samples at these time points. The sample was reconstituted in organic solvents such as 50/50) of 70% (v/v) formic acid and 50% (v/v) prior to analysis. Although this is being done for MALDI analysis, the potential effects of these solvents on the samples can be studied.
  2. I see the samples from healthy individuals were collected and the number has declined on D10 as described in the sentence “negative at the enrollment day (D0, n=51) and ten days later (D10, n=40) (Table 1)”. If this is correct, the number of healthy individuals has been declined at D10 and if these samples have been considered for the building the training model could mislead for the validation. This should be clarified and stated in the result section.
  3. According to the sentence 234 and 235, 32 were considered exempted from infection at D30, can authors explain what about remaining cases? Did these individuals excluded from final analysis and why they were exempted?
  4. Training data set is very small to train ML methods with cross validation and questions the reliability of the ML model built to use in real time lab diagnosis with many variables of platform, storage conditions, etc.
  5. Training data set is very small to train ML methods with cross validation and questions the reliability of the ML model built so to use in real time lab diagnosis with many variables of platform, storage conditions, etc.
  6. Training data set is very small to train ML methods with cross validation and questions the reliability of the ML model built so to use in real time lab diagnosis with many variables of platform, storage conditions, etc.
  7. The patient information such viral load determined from qRT-PCR tests can be used to correlate with MS spectra. Authors can comment on how the intensity of the spectra matches in saliva samples compared with NPS of the same individual.
  8. Total 9 most intense peaks were used to differentiate between healthy and COVID positive samples. It would be interesting to see if these peaks are is of proteins derived from humans or virus.
  9. Authors have used five different tests for the prediction of positive and negative cases. But it is unclear from the results which one was used for final analysis and how sensitivity and specificity was calculated. Is it average of the all five tests or outcome of the single test?
  10. Another article (PMID: 34240163) by one of the groups has used machine learning approach for the COVID-19 diagnosis. This article can be discussed under the discussion section.